# How Powerful are Graph Neural Networks with Random Weights?

## Abstract

Thanks to the great success of graph neural networks (GNNs) in structural information learning, extensive variants by virtue of sampling or pooling have been developed to further improve the performance, scalability, and applicability. However, there is still room for improvement in learning efficiency because current GNNs are trained via batch gradient descent with many graphs in each iteration. The good potential of random features in speeding up the training phase motivates us to consider the expressive power of GNNs with random weights. Based on the framework of Graph Isomorphism Network, we propose a novel model called Hashing Graph Isomorphism Network (HashGIN) with only one epoch of training by revising the convolutional layer with random hash functions and adjusting the learning objective with regularized least squares loss. In light of the property of $k$ random hash functions, we theoretically show that the injective phase in the Weisfeiler-Lehman test can be approximated by a hash family. An approximation upper bound is further provided with rigorous mathematical proof for the convergence of our model. Our experiments on several benchmark datasets show that HashGIN is effective and efficient for graph classification tasks. Compared to the state-of-the-art methods, HashGIN achieves better or comparable accuracies with less training time and memory cost.

## 1 Introduction

In recent years, deep learning has made significant progress in various fields, such as natural language processing (Floridi & Chiriatti, 2020) and computer vision (He et al., 2016; Redmon et al., 2016), opening up new possibilities in each domain. With the explosive growth of internet data, the amount of data available for training models has also been increasing. Deep learning enables automatic learning from large-scale data through gradient descent. Moreover, in order to improve model performance, increasingly sophisticated and complex models have been developed. For instance, large language models like ChatGPT (Ouyang et al., 2022) have seen a steady increase in the number of parameters, demanding substantial computational resources for model training. This has prompted researchers to explore techniques to overcome the challenges posed by the growing complexity and resource requirements in deep learning.

In addition to the aforementioned techniques that prioritize data with a grid structure, graphs stand out as a highly expressive and versatile data structure in many branches of deep learning, such as social networks, transportation networks, computer networks, knowledge graphs, and recommendation systems. Graph neural networks, as an emerging deep learning model, have attracted considerable attention from both academia and industry due to their powerful ability to deal with graph data. GNNs enable end-to-end learning on graph structures by integrating information from nodes and edges, effectively uncovering latent patterns and features in graph data. There have been remarkable progress and achievements in various domains, including social network analysis (Backstrom & Leskovec, 2011), recommendation systems (He et al., 2020), and chemical computation (Coley et al., 2019), showcasing exemplary performance.

In general, graph-related tasks can be roughly viewed as node classification, link prediction, and graph classification on graphs. In this paper, we focus on the graph classification task [1], where

---

[1] For detailed information on related work, please refer to the appendix.

many popular GNN architectures can automatically extract structural features of graphs through recursive message passing, such as GraphSAGE (Hamilton et al., 2017), DGCNN (Zhang et al., 2018) and GIN (Xu et al., 2019). These elaborate GNN structures can be considered as variants of the Weisfeiler-Lehman (WL) graph isomorphism test (Leman & Weisfeiler, 1968; Xu et al., 2019), which is a widely recognized algorithm used for determining graph isomorphism. The algorithm iteratively labels nodes based on their own labels and the labels of neighboring nodes, leveraging defined rules for label aggregation and updating. GIN has been proven to exhibit the same level of power as the Weisfeiler-Lehman graph isomorphism test in theory (Xu et al., 2019). In order to further improve the accuracy of graph classification, models such as DiffPool (Ying et al., 2018) and GMT (Baek et al., 2021) have been proposed to explore the graph pooling scheme. Alternatively, KPGNN (Feng et al., 2022) improves $k$-hop messaging to leverage more graph structure information in each hop. Despite the enhanced accuracy in graph classification achieved by these ways, it comes at the cost of increased computing resources and training time consumption.

Our objective is to discover an approach that can accelerate the training process of GNNs while maintaining a comparable level of performance. Fortunately, we can achieve this using neural network techniques with random parameters. In a probabilistic sense, a neural network with fixed random hidden parameters, obtained through the design of a suitable distribution, has the potential for universal approximation capabilities (Igelnik & Pao, 1995; Husmeier & Husmeier, 1999b). The integration of neural network techniques with random parameters eliminates the need for costly gradient descent training methods. Neural network techniques with random parameters have much lower training complexity compared to the traditional training of back propagation method.

In this paper, we are the first to apply graph neural networks with random parameters for graph classification tasks. Only one training epoch is required, greatly accelerating the training speed. To achieve this, we have designed $k$ random hash functions for the convolutional layer to efficiently differentiate between different neighboring node features. We have theoretically proven that the $k$ random hash functions can achieve the same power as the WL test. Moreover, we adopt the mean squared loss function as the learning objective to obtain the parameters of the classifier. We name our proposed model Hashing Graph Isomorphism Network (HashGIN). We present a meticulous mathematical proof that examines the convergence characteristics of the model and its upper bound. To validate the performance of HashGIN, we conducted extensive experiments using various graph classification datasets. The results demonstrate that HashGIN achieves comparable classification performance to the SOTA models across multiple datasets. Notably, HashGIN surpasses the competition in terms of time and space efficiency, establishing its superiority. Furthermore, we conduct ablation experiments to assess the effectiveness of the $k$ random hash functions. Our contributions are summarized as follows:

- By utilizing the $k$ random hash functions, we propose HashGIN to achieve fast graph convolution with only one epoch learning for graph classification.

- We prove that the $k$ random hash functions can achieve performance similar to the WL testing in theory. Additionally, we provide a rigorous mathematical proof of the approximate upper bound on model convergence.

- Expensive experiments have shown that our model exhibits extremely high training efficiency, while achieving superior classification performance.

## 2 PRELIMINARYS

**Problem formulation.** A graph can be represented as $G = \{V, \mathcal{E}, X\}$, where $V = \{1, 2, \cdots, n\}$ is the node set and $\mathcal{E} \subseteq V \times V$ is the edge set of the graph. Correspondingly, $A = \{0, 1\}^{n \times n}$ is the adjacency matrix. For $v \in [1, n]$, let $x_v \in \mathbb{R}^d$ denote the feature vector of the node, and let $X \in \mathbb{R}^{n \times d}$ represent the ensemble of the feature vectors of all nodes. In the graph classification task, there are $N$ graphs and $\mathcal{Y}$ denotes the label set of graphs. $y_i \in \mathcal{Y}$ is the label of the $i$-th graph. The task is formulated as predicting the label using the node features and structural information.

**GNNs.** In general, GNNs can be reduced to learning node representations by recursively aggregating and combining neighbor node information as follows:

$$a_v^{(l)} = \text{AGGREGATE}^{(l)}\big(\{h_u^{l-1} : u \in \mathcal{N}(v)\}\big), \quad h_v^{(l)} = \text{COMBINE}^{(l)}\big(h_v^{(l-1)}, a_v^{(l)}\big), \qquad (1)$$

where $h_v^{(l)}$ is the node representation of the $l$-th layer for $l = 1, 2, \ldots, L$. $\mathcal{N}(v)$ is the set of neighbors of node $v$. For graph-level tasks, a readout function is used in conjunction with graph-level pooling to aggregate information from all nodes:

$$h_G = \text{READOUT}\big(\{h_v^{(L)}\}|v \in V\big). \tag{2}$$

Here the $\text{AGGREGATE}^{(l)}(\cdot)$ and $\text{READOUT}(\cdot)$ functions are both invariant functions. GIN is a representative model that follows this approach and is considered as powerful as the WL test in theory (Xu et al., 2019). It can be formulated as:

$$h_v^{(l)} = \phi\bigg( h_v^{(l-1)}, f\Big(\big\{ h_u^{(l-1)} : u \in \mathcal{N}(v) \big\}\Big) \bigg), \tag{3}$$

where $f$ is the function that operates on a multiset. Xu et al. (2019) proved that if $\phi$ and $f$ used in a GNN are injective, GNN is capable of distinguishing any pair of non-isomorphic graphs $G_1$ and $G_2$ in the WL test. Specifically, GIN uses multilayer perceptrons (MLPs) to approximate $f \circ \phi$:

$$h_v^{(l)} = \text{MLP}^{(l)}\bigg( \Big(1 + \epsilon^{(l)}\Big) \cdot h_v^{(l-1)} + \sum_{u \in \mathcal{N}(v)} h_u^{(l-1)} \bigg), \tag{4}$$

where $\epsilon$ is a learnable parameter or a constant. Initially, $h_v^{(0)} = x_v$ as the feature vector of node $v$.

## 3 METHODOLOGY

In this section, we design a hash function to achieve efficient aggregation of neighborhood information and propose a graph neural network with random weights. At the same time, we theoretically prove that our method is expected to be as powerful as the WL test. We further provide a rigorous mathematical proof for the convergence of our model, along with an approximate upper bound.

### 3.1 HASHGIN

In GIN, the MLPs are used to implement the function $f \circ \phi$. MLPs in each convolution layer lead to a large computational overhead in the training phase. In our approach, the function $f \circ \phi$ is implemented with the $k$ random hash function:

$$h_v^{(l)} = k\text{-HASH}^{(l)}(S), \quad S = \Big\{ h_u^{(l-1)} : u \in \mathcal{N}(v) \cup \{v\} \Big\}, \tag{5}$$

$$k\text{-HASH}^{(l)}(S) = \sum_{h \in S} CONCAT_{k=1}^{K} hash_k^{(l)}(h), \tag{6}$$

where $k$-HASH consists of $K$ random hash functions. The $k$-HASH$^{(l)}$ aims to implement the $f \circ \phi$ and replicate the multiset injection implemented by MLPs in GIN. For $l = 0$, we initialize the node feature vectors as $h_v^{(0)} = x_v$. After $L$ GNN layers, we obtain the final representation of the nodes. The structural information and node features of each graph are integrated into the multiset of node representations. The READOUT function is necessary for obtaining the full graph representation. We perform the READOUT function by $k$-HASH on the multiset of node representations:

$$h_G = k\text{-HASH}\Big(\Big\{ h_v^{(L)} : v \in V \Big\}\Big). \tag{7}$$

Finally, $H = [H_{G_1}, H_{G_2}, \ldots, H_{G_N}]^T \in \mathbb{R}^{N \times D}$ represents all the graphs, where $N$ is the number of graphs and $D$ represents the output dimension. We treat classification tasks as the regularized least squares minimization problem:

$$\min_{\beta} \|H\beta - Y\|^2 + \lambda \|\beta\|_2^2, \tag{8}$$

where $\beta \in \mathbb{R}^{D \times |\mathcal{Y}|}$ is the the output weight matrix. $Y \in \mathbb{R}^{N \times |\mathcal{Y}|}$ is the one-hot matrix where $Y_{i,y_i} = 1$. The analytical solution of the problem is:

$$\beta = \begin{cases} (H^T H + \lambda I)^{-1} H^T Y, N \geqslant D \\ H^T (HH^T + \lambda I)^{-1} Y, N < D, \end{cases} \tag{9}$$

where $\lambda$ is the regularization parameter.

---

**Algorithm 1** Hashing Graph Isomorphism Network

---

**Input**: Train set size $N$ and test set size $M$, adjacency matrix list $A = [A_1, A_2, \cdots, A_{N+M}]$, node feature matrix list $H = [H_1, H_2, \cdots, H_{N+M}]$, graph label one-hot matrix of train dataset $Y$
**Parameter**: Number of layer $L$, number of hash $K$, hidden dimension $D$, regularization parameter $\lambda$
**Output**: Predicted label of the test set

 1: **function** $GraphConvolution(A,H)$
 2:     $n \leftarrow length(A)$
 3:     **for** $i \leftarrow 1$ to $n$ **do**
 4:         $H_i^{(0)} \leftarrow H_i$
 5:         **for** $l \leftarrow 1$ to $L$ **do**
                // correspond to Equation 6
 6:             $H_i^{(l-1)'} \leftarrow CONCAT_{k=1}^{K} H_i^{(l-1)} W_k^{(l-1)} + b_k^{(l-1)}$
 7:             $H_i^{(l)} \leftarrow A_i H_i^{(l-1)'}$
 8:         **end for**
 9:         $H_i^{(L)'} \leftarrow CONCAT_{k=1}^{K} H_i^{(L)} W_k^{(L)} + b_k^{(L)}$
10:         $H_{G_i} \leftarrow \sum H_i^{(L)'}$
11:     **end for**
12:     **return** $H_G$
13: **end function**
14: Detach $A^{train}, A^{test}$ from $A$ and $H^{train}, H^{test}$ from $H$
15: **for** $l \leftarrow 0$ to $L$ **do**
16:     **for** $k \leftarrow 1$ to $k$ **do**
17:         Initialize $W_k^{(l)}, b_k^{(l)}$ from the uniform distribution
18:     **end for**
19: **end for**
20: $H_G^{train} \leftarrow GraphConvolution(A^{train}, H^{train})$
21: $H_G^{test} \leftarrow GraphConvolution(A^{test}, H^{test})$
22: **if** $N \geq D$ **then**
23:     $\beta \leftarrow (H_G^{trainT} H_G^{train} + \lambda I)^{-1} H_G^{trainT} Y$
24: **else**
25:     $\beta \leftarrow H_G^{trainT} (H_G^{train} H_G^{trainT} + \lambda I)^{-1} Y$
26: **end if**
27: $Z \leftarrow H_G^{test} \beta$
28: **return** $argmax(Z_i)$

---

## 3.2 ALGORITHM IMPLEMENTATION

In this section, we will introduce the implementation of HashGIN, which requires only one training epoch and is highly efficient. To achieve this, we propose the $k$-HASH applied to the convolution layer. The $k$-HASH is composed of $K$ random hash functions, where the $k$-th hash function is defined as follows:

$$hash_k^{(l)}\left(h_v^{(l)}\right) = \sigma\left(h_v^{(l)} W_k^{(l)} + b_k^{(l)}\right), \tag{10}$$

where $W_k^{(l)} \in \mathbb{R}^{D \times \frac{D}{K}}, b_k^{(l)} \in \mathbb{R}^{\frac{D}{K}}$, $W$ and $b$ are initialized from the uniform distribution $(-\theta, \theta)$, and $D$ is the hidden dimension. In order to make the hash function non-linear, $\sigma$ is the activation function that can be selected as ReLU or sigmoid function. In the final classification, the parameters of $\beta$ are derived using Equation 9 on the training set. Later, $\beta$ is applied to classify the graphs:

$$Z = H\beta, \tag{11}$$

where $Z \in \mathbb{R}^{N \times |\mathcal{Y}|}$ and $argmax(Z_i)$ is the prediction result of the $i$-th graph. A running example of HashGIN is illustrated in Algorithm 1.

**Time complexity.** Based on this comprehensive time complexity analysis, we are going to demonstrate the high efficiency of our proposed model. For a single graph, the overall time complexity of the model is $\mathcal{O}(|E|d + (L-1)|E|D + LD^2 + D|\mathcal{Y}|)$, where $d$ and $D$ represent the dimensions of the node features and the hidden embedding, respectively, and $L$ denotes the number of GNN layer. In particular, each $hash_i$ in $k$-HASH requires $\mathcal{O}(D^2/K)$. Therefore, the time complexity of $k$-HASH is

$\mathcal{O}(D^2)$. Finally, the time complexity of $\mathcal{O}(D|\mathcal{Y}|)$ is required after obtaining the graph representation in order to predict the category using $\beta$. The $\beta$ is calculated from the training set using Equation 9, which has a time complexity of $\mathcal{O}(N_{tr}D^2)$. Here, $N_{tr}$ represents the size of the training set.

**Remark.** The high efficiency of our model is due to the fact that the parameters in HashGIN are obtained through one epoch calculation instead of being trained using gradient descent. Compared to GIN, we design the $k$-HASH function as the injective function on the multiset instead of MLPs. In addition, we predict the class of the graph by minimizing the regularized least squares, where only one computation is performed using the representations and labels of the graphs in the training set. Consequently, our model exhibits a low computational and memory overhead.

### 3.3 THEORETICAL JUSTIFICATION

In this section, we aim to explain the reasons why HashGIN can function effectively. Specifically, we first provide a theoretical analysis of the injective approximation to demonstrate that the capability of HashGIN is as powerful as the WL test (or GIN). Then, we further theoretically reveal that the algorithm implementation converges under certain conditions.

**Injective approximation via $k$ random hash functions.** In the theoretical justification of GIN (Xu et al., 2019), GNN can be as powerful as the WL test when the message-passing and readout functions are injective. Here, we follow the design of GIN to show why the neighborhood aggregation of HashGIN is injective.

**Theorem 1.** *Let $\mathcal{X}$ be the input feature space and assume $\mathcal{X}$ is countable. There exists a function $\mathcal{H} : \mathcal{X} \to \mathbb{R}^d$ so that $\mathcal{H}(X) = k$-HASH$(X)$ is unique for each multiset $X \subset \mathcal{X}$ of bounded size.*

*Proof.* The injectiveness of multiset $X \subset \mathcal{X}$ is a super problem of *"set membership problem"*. [2] The set membership problem is defined as follows. Let $[m]$ be the universe, and let the input be $S \subset [m]$ with $|S| = n$. Based on $S$, a data structure $f$ can be used to reconstruct $S$, and, therefore, must distinguish between all size $n$ subsets of $[m]$. To address the set membership problem, the data structure makes crucial use of 2-wise independent hash functions. In particular, let $\mathcal{H} = \{h : [m] \to [l]\}$ be a 2-wise hash family and let $l \geq n^2$. Then for any $S \subset [m]$ of size $|S| = n$, there exists an $h \in \mathcal{H}$ such that the restriction $h|_S$ is a one to one function from $S$ to $[l]$. Based on the hash family to identify every $S$ with $n$, we can identify every $S$ with any size via identify the size of $S$ first. Furthermore, to further reduce the risk of total collisions, $k$ hash functions are adopted due to the larger range $\mathcal{H} : \mathcal{X} \to \{1, \ldots, l^k\}$. Thus, $\mathcal{H}(X) = k$-HASH$(X)$ is an injective function of multisets. $\square$

**Remark.** In proof of injectiveness in GIN (i.e., the proof of Lemma 5), there exists a compressed form of an one-hot embedding that makes $h(X) = \sum_{x \in X} f(x)$ is an injective function of multisets. In practice, GIN adopts the MLPs to model and learn $f$ because of the universal approximation theorem (Hornik et al., 1989; Hornik, 1991). In the implementation of HashGIN, considering the excellent performance of $k$ random hash functions in tackling collisions, the method proposed in this article is an extension of the WL subtree kernel where we use $k$ random hash functions instead of a single hash function. With the mechanism for modeling universal multiset functions in Theorem 1 as a building block, we can conceive aggregation schemes that can represent universal functions over a node and the multiset of its neighbors, and thus will satisfy the injectiveness condition required by Theorem 3 of GIN.

**Hashing theory.** Let $h : \mathcal{X} \to \{0, \ldots, \mathcal{T}\}$ be a hash function in $k$-HASH family. Then the probability $p_{col}$ that $g \in \mathcal{X}$ collides with one or more other input features is given by

$$p_{col} = 1 - (1 - \frac{1}{\mathcal{T}})^{|\mathcal{X}|-1} \tag{12}$$

For large $\mathcal{T}$, we have the approximation

$$p_{col} \approx 1 - e^{-\frac{|\mathcal{X}|}{\mathcal{T}}}. \tag{13}$$

The expected number of features in collision $C_{gog}$ is given by

$$C_{gog} = |\mathcal{X}|p_{col} \tag{14}$$

---

[2] A detailed description of set membership problem is shown in the supplementary materials.

The combination of multiple hash functions approximates a single hash function with much larger range $h : \mathcal{X} \to \{1, \dots, \mathcal{T}^K\}$, which drastically reduces the risk of total collisions:

$$p_{col} \approx 1 - e^{-\frac{|\mathcal{X}|}{\mathcal{T}^K}}. \tag{15}$$

Namely, the probability that any two features in $\mathcal{X}$ will be mapped by $k$-HASH function to the same element is $1 - e^{-\frac{|\mathcal{X}|}{\mathcal{T}^k}}$. In other word, the probability that $k$-HASH function is injective is $e^{-\frac{|\mathcal{X}|}{\mathcal{T}^k}}$. This probability will sharply increase with the dimensions $T$ of the mapping space chosen by our random hash function and the number of random hash functions $K$.

**Remark.** According to Equation 15, it is not possible to completely avoid total collisions. In fact, the probability of conflict described in Equation 15 indicates that the space of $X$ has a linear impact, while the size of $K$ has an exponential impact, particularly when considering the space of actual hidden representations. As a result, increasing $K$ by just one unit can significantly expand our mapping space, allowing us to effectively simulate the Weisfeiler-Lehman (WL) test.

**Upper bound for approximation.** In practice, each HashGIN model can be viewed as a random variable in a certain probability space, and its approximation capability is probabilistic rather than deterministic, similar to that of RVFL networks as studied in (Igelnik & Pao, 1995; Husmeier & Husmeier, 1999a), random features discussed in (Rahimi & Recht, 2008).

Let $f : V \to R$, be a bounded and continuous function defined on a vertex set $V$, of which each vertex has $m$ features. According to the setting of graph classification, the ideal function of graph classification that our model wants to approximate is actually the summation of these functions. At the same time, since the dataset we input is a finite set, we only need to find a way to prove that the $f$ function has a finite upper bound. $g$ is the activation function satisfying $|g(x)| \le 1$ and $\int_K g'(t)dt < \infty$ (for any compact set $K$) (Adams & Fournier, 2003). Denote $\hat{A}^k_{\mu,v} := \hat{A}^k(\mu, \nu)$ as the $(\mu, \nu)$ element of a given matrix $\hat{A}^k$, where $k$ is a positive integer. $x_\mu, x_v$ stand for the feature vector of vertex $\mu$ and vertex $v$, respectively. Based on the integral representation framework (Igelnik & Pao, 1995; Murata, 1996), the unknown function can be formulated as

$$f(x_v) = \int_{R^m} \alpha(w) g \left( w^{\mathrm{T}} \sum_{\mu \in \mathcal{N}(\nu)} \hat{A}^k_{\mu,v} x_\mu \right) dw, \tag{16}$$

where $\alpha : R^m \to R$ is the (continuous) functional representation of the output weights, as discussed later in this section. Theoretically, the above integral representation can be approximated in the sense that (Igelnik & Pao, 1995; Murata, 1996).

$$f(x_v) = \lim_{\theta \to +\infty} \int_{-\theta}^{\theta} \cdots \int_{-\theta}^{\theta} \alpha(w_1, \dots, w_m) \cdot g \left( w^{\mathrm{T}} \sum_{\mu \in \mathcal{N}(v)} \hat{A}^k_{\mu,v} x_\mu \right) dw_1 \dots dw_m. \tag{17}$$

For simplicity, we denote

$$f_\theta(x_v) := \int_\Theta \alpha(w) g \left( w^T \sum_{\mu \in \mathcal{N}(v)} \hat{A}^k_{\mu,v} x_\mu \right) dw, \tag{18}$$

with $\Theta := [-\theta, \theta]^m, |\Theta| = (2\theta)^m$.

The estimation of $f_\theta(x_v)$ can be obtained using the Monte-Carlo method, that is,

$$\hat{f}_L(x_v) = \frac{|\Theta|}{L} \sum_{j=1}^{L} \alpha(w_j) g \left( w_j^{\mathrm{T}} \sum_{\mu \in \mathcal{N}(v)} \hat{A}^k_{\mu,v} x_\mu \right), \tag{19}$$

where $\{w_1, w_2, \dots, w_L\}$ is a sample of size $L$ selected randomly (and independently) from the uniform distribution with the probability measure:

$$P(w) = \begin{cases} |\Theta|^{-1} & \text{if } w_j \in \Theta \\ 0 & \text{otherwise} \end{cases}. \tag{20}$$

Mathematically, $\hat{f}_L$ has a similar functional form as the proposed HashGIN model, that is

$$\hat{f}_L(x_v) = \sum_{j=1}^{L} \beta_j g\left(w_j^{\mathrm{T}} \sum_{\mu \in \mathcal{N}(v)} \hat{A}_{\mu,v}^k x_\mu\right),\tag{21}$$

where $\beta_j := \alpha(w_j)|\Theta|/L$. Given the graph with node feature matrix $X \in R^{N \times m}$ and the renormalized adjacency matrix $\hat{A} \in R^{N \times N}$, the discrete representation of Equation 21 in matrix form is exactly the same as Equation 20. Denote the distance between f and $\hat{f}_L$ by

$$d_V\left(f, \hat{f}_L\right) := \sqrt{\frac{1}{|V|}E\left[\sum_v \left(f(x_v) - \hat{f}_L(x_v)\right)^2\right]},\tag{22}$$

where $E[\cdot]$ denotes the expectation value with respect to the probability distribution of $w$ (see Equation 20), and $|V|$ denotes the volume of the vertex set $V$.

With these notations, we can generally formulate the convergence property of HashGIN as follows.

**Theorem 2.** *There exists certain uniform distribution $\Theta := [-\theta, \theta]^m$, for a HashGIN model with sufficiently large L, its approximation upper bound is characterized in the probability sense that*

$$d_V\left(f, \hat{f}_L\right) \le \frac{C}{\sqrt{L}}, C := \sqrt{|2\theta|^m \int_\Theta \alpha^2(w)dw}$$

The upper bound in Theorem 2 depicts that there exist certain appropriate distributions, which highly depend on the functional class and complexity of the function to be approximated, making it feasible that a HashGIN model (with its random weights uniformly drawn from that distribution) can approximate a given bounded and continuous function defined on the graph with high probability (when $L$ is sufficiently large). Due to page limitation, we provide the detailed proof of Theorem 2 in the supplemental materials.

## 4 EXPERIMENTS

In this section, we conduct several experiments on graph classification to demonstrate that HashGIN can achieve performance comparable to SOTA GNNs. Meanwhile, it is worth highlighting that our model boasts a significantly low computational and memory cost.

### 4.1 EXPERIMENTAL SETTING

**Datasets.** The datasets are selected from TU datasets (Morris et al., 2020), including social datasets and bioinformatics datasets, as well as binary and multi-class classification datasets. In detail, IMDB-BINARY, IMDB-MULTI, COLLAB are social datasets and PROTEINS, D&D, NCI1 are bioinformatics datasets. Graphs in biological datasets have initial node features. At the same time, there is no node information in the social dataset so the structural information of the graph is the only information that the model can capture for prediction. Dataset statistics are reported in Table 1.

**Configurations.** Following the evaluation protocol for graph classification proposed by Errica et al. (2019), we perform 10-fold cross-validation evaluation and divide the data into training/validation/test sets, in which 10 percent of the training set is selected as the validation set. For hyperparameters, the $\theta$ initializing the hash is fixed to 1. We adjust $K$ in $\{2^0, 2^1, 2^2, 2^3, 2^4, 2^5, 2^6\}$ for $k$-HASH. The $\lambda$ is searched in $\{1,2,4,8,10,16,20\}$ and the hidden size is set to 128. We search the number of GNN layers in $\{1, \cdots, 5\}$. Then the $\beta$ is calculated using the graph representations and labels from the training set. Thus, there is no need to go through multiple epochs of training.

**Baselines.** In order to verify that HashGIN accelerates training a lot compared to message propagation framework, we conducted a comparison with several widely-used GNNs instead of the current SOTA models. Following the experimental setting, the adopted baselines from (Errica et al., 2019) includes: (1) DGCNN (Zhang et al., 2018) proposes an end-to-end training approach for graph. (2) DiffPool (Ying et al., 2018) is able to capture hierarchical structure in graphs and has shown improved

Table 1: The overall performance comparison for graph classification. The results of baselines are derived from Errica et al. (2019). OOR (Out of Resources) indicates that the training time is too long (over 72 hours for a single training) or GPU memory.

| | Social Network | | | Bioinformatics | | |
|---|---|---|---|---|---|---|
| | IMDB-BINARY | IMDB-MULTI | COLLAB | PROTEINS | D&D | NCI1 |
| # Graphs | 1,000 | 1,500 | 5,000 | 1,113 | 1,178 | 4,110 |
| # Classes | 2 | 3 | 3 | 2 | 2 | 2 |
| Avg. # Nodes | 19.8 | 13.0 | 74.5 | 39.1 | 284.3 | 29.8 |
| DGCNN | 69.2±3.0 | 45.6±3.4 | 71.2±1.9 | 72.9±3.5 | 75.6±4.3 | 76.4±1.7 |
| DiffPool | 68.4±3.3 | 45.6±3.4 | 68.9±2.0 | 73.7±3.5 | 75.0±3.5 | 76.9±1.9 |
| ECC | 67.7±2.8 | 43.5±3.1 | OOR | 72.3±3.4 | 72.6±4.1 | 76.2±1.4 |
| GraphSAGE | 68.8±4.5 | 47.6±3.5 | 73.9±1.7 | 73.0±4.5 | 72.9±2.0 | 76.0±1.8 |
| GIN | 71.2±3.9 | 48.5±3.3 | 75.6±2.3 | 73.0±4.5 | 75.3±2.9 | **80.0±1.4** |
| **HashGIN** | **72.9±0.4** | **49.2±0.9** | **79.9±0.3** | **76.2±2.0** | **80.6±1.3** | 78.1±0.9 |

Table 2: Training time and GPU memory comparison on all datasets.

| | Social Network | | | Bioinformatics | | |
|---|---|---|---|---|---|---|
| | IMDB-BINARY | IMDB-MULTI | COLLAB | PROTEINS | D&D | NCI1 |
| GIN | 4m39s | 9m32s | 18m18s | 8m44s | 24m56s | 6m24s |
| **HashGIN (Full Batch)** | **24s** | **37s** | **11m48s** | **22s** | **48s** | **3m0s** |
| **HashGIN (mini Batch)** | 31s | 50s | 14m37s | 36s | 61s | 6m02s |
| GIN | 697.6 MB | 693.4 MB | 825.7 MB | 749.1 MB | 818.8MB | 703.4MB |
| **HashGIN (Full Batch)** | 75.9 MB | 76.8 MB | 4292.9 MB | 83.2 MB | 488.8 MB | 150.8 MB |
| **HashGIN (Mini Batch)** | **9.7 MB** | **6.6 MB** | **109.9 MB** | **9.6 MB** | **53.1 MB** | **4.7 MB** |

performance over other GNN architectures. (3) ECC (Simonovsky & Komodakis, 2017) extend the convolution operator from regular grids to arbitrary graphs without utilizing the spectral domain. (4) GIN (Xu et al., 2019) is a extremely powerful GNN using a message passing scheme to propagate information between nodes. (5) GraphSAGE (Hamilton et al., 2017) aims to learn node representations by aggregating information from the node's fixed-size neighbors.

## 4.2 PERFORMANCE COMPARISON

We conducted experiments to validate both the effectiveness and efficiency of HashGIN. In order to assess the efficiency of HashGIN, we compare its training time and GPU memory cost against the comparative GIN.

**Effectiveness comparison.** Table 1 illustrates a comparative evaluation of classification accuracy between HashGIN and other methods. Overall, we can observe that HashGIN achieves the best performance on most benchmarks, which implies the effectiveness of HashGIN in structural information encoding. In particular, except for NCI1, HashGIN surpasses GIN on five out of six datasets, signifying the ability of our custom $k$-HASH function in aggregating neighboring information. Based on the experimental accuracies presented above, we can conclude that HashGIN is capable of differentiating graph structures via random weights.

**Efficiency comparison.** Based on the randomization technique, our designed $k$-HASH function can approximate the function of an injective function on multisets, which greatly accelerates the training speed of the convolutional layers in GNNs and has lower memory usage. To verify this, we conduct experiments to evaluate the cost of time and memory of one of 10-fold cross-validation evaluation. In addition, we verify the efficiency at random graphs of different sizes, where the number of nodes is $\{100, 200, \ldots, 1000\}$ and the average node degree is 5. The experimental results are shown in Table 2 and Figure 1, where each experiment uses 36 cores Intel Golden 6240 as well as 1 NVIDIA V100 GPU. It is worth noting that HashGIN has the capability to conduct full batch training, whereas the majority of deep learning models today utilize minibatch training. To better compare, we also measure the memory usage of HashGIN during minibatch training, with the batch size kept the same as GIN.

We can find the computational and memory efficiency of HashGIN is remarkably excellent. The parameters in the convolutional layers are obtained through well-designed function modeling and random strategies. Conversely, the parameters in the final prediction layer are determined by computing the objective of minimizing the squared loss, which is evaluated using the training set. During the entire training process, all the data is looked only once. Therefor, HashGIN has excellent computational efficiency.

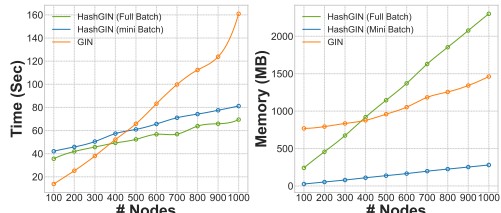

Figure 1: Efficiency comparison on different graph sizes.

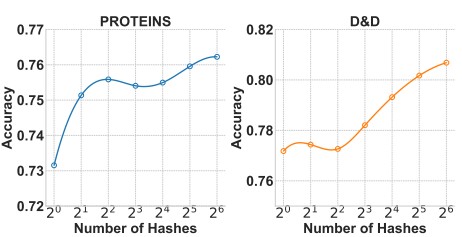

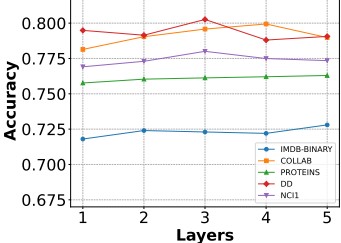

Figure 2: The impact of the number of hashes.

Figure 3: The impact of the number of layers.

### 4.3 ABLATION STUDY

To highlight the effectiveness of the $k$-HASH, we conducted ablation experiments by systematically adjusting the number of hashes while keeping other parameters consistent. We ensured that the sum of the output vector lengths of the $k$-HASH functions remained constant, maintaining a fair comparison of model parameters. Our experiments show that a smaller number of hashes adversely affects model performance due to higher collision rates, hindering accurate identification of neighboring structure information. Conversely, increasing the number of hashes improves classification accuracy, highlighting the efficacy of combining multiple hashes to reduce hash conflicts. The results validate the advantages of the $k$-HASH.

In addition, we conduct a series of experiments to assess the influence of the number of GNN layer on classification accuracy. We perform layer adjustments within the range of $\{1, 2, 3, 4, 5\}$, while keeping other parameters fixed. Considering the varying sizes and densities of distinct dataset graphs, it becomes evident that each dataset possesses its own optimal number of layers. Fewer layers hardly achieve the best performance on these datasets. It shows the efficacy of HashGIN in capturing the structural information of the graph through the utilization of multiple graph convolution layers.

## 5 CONCLUSIONS

In this paper, we aim to investigate the feasibility of graph neural network with random weights for graph classification. By utilizing the $k$ random hash functions in the convolutional layer and adapting the learning objective to regularized least square, we propose HashGIN, a model that achieves competitive accuracies in graph classification tasks while significantly reducing training time and memory usage. Further theoretical analysis conducted in this study demonstrates the approximability of the injective phase in the WL test using $k$-HASH functions. We also provide a rigorous mathematical proof for the convergence of our model, supporting the effectiveness of HashGIN in capturing and leveraging graph structure information. Experimental results confirm the efficacy and efficiency of HashGIN, showing competitive performance while improving training time and memory consumption. In conclusion, HashGIN offers a promising solution for improving GNN efficiency using random weights.

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

## A    RELATED WORK

**Graph Neural Networks.** In recent years, GNNs have gained significant attention in the field of machine learning and graph analysis. Several studies have explored different approaches and architectures to effectively model and learn from graph-structured data. Kipf & Welling (2016) proposed GCN, which is a spectral-based convolutional layer that applies graph Fourier transforms to the graph Laplacian. Based on spectral graph theroy, SSGC (Zhu & Koniusz, 2020) extends markov diffusion kernel and utilizes normalized Laplacian regularization to generate feature maps. In the graph classification task, many GNNs are designed based on the message propagation framework. Through iterative message passing, GNNs update the feature vectors of nodes by aggregating information from their neighboring nodes. There are elaborate graph neural network based on the framework, such as GraphSAGE (Hamilton et al., 2017), DGCNN (Zhang et al., 2018), and GIN (Xu et al., 2019). The theoretical analysis has shown that GIN exhibits same expressive power to the WL test (Xu et al., 2019). In the WL test, a iterative procedure is employed to assign labels to nodes, taking into account both their own labels and the labels of neighboring nodes. This process incorporates rules for label aggregation and updating. By utilizing its iterative approach, the WL test effectively captures the structural characteristics of graphs and facilitates precise evaluation of isomorphism by comparing the resulting label sets.

**Neural Networks with Random Parameters.** Several related studies have explored the concept of randomized neural networks. Randomly assigning a subset of the network's weights, which transforms the optimization task into a linear least-squares problem. This technique has been successfully applied to both feedforward (Pao et al., 1994) and recurrent networks (Lukoševičius & Jaeger, 2009). A key challenge in this field is how to effectively assign random input weights and biases while ensuring that the resulting model possesses universal approximation capabilities. RVFL (Husmeier & Husmeier, 1999b) involves randomly assigning input weights and biases according to a predetermined probability distribution. This straightforward method has been widely used. In order to achieve universal approximation capabilities in a probabilistic sense, the choice of distribution is crucial and a uniform range of $[-\lambda, \lambda]$ with $\lambda > 0$ is often selected. To achieve the potential of universal approximation in neural network, it is essential to design a suitable distribution for obtaining random hidden parameters. Inspired by this challenge, stochastic configuration networks (SCNs) (Wang & Li, 2017) and subsequent extensions (Wang & Li, 2018; Li & Wang, 2019) are introduced. SCNs address the fundamental task of constructing a universal approximator using random basis functions by incrementally and stochastically configuring input weights and biases. As more hidden nodes are added, the model's universal approximation capabilities improve. SCNs are the first attempts to adaptively determine the optimal support range for building a universal approximator using random basis functions. GCN-RW first introduces randomized learning techniques to extend GCN. It modifies the convolutional layer by incorporating random filters and speeds up the training of GCNs. The effectiveness and efficiency of GCN-RW are empirically demonstrated through its successful application in semi-supervised node classification tasks. Despite its promising applications in node classification, randomness in neural networks have not been extensively explored in the context of graph classification tasks. Therefore, our work aims to bridge this gap and investigate the effectiveness in graph classification tasks.

## B    SET MEMBERSHIP PROBLEM

In this section, we are going to introduce the set membership problem, including data structures and query algorithms. Here, we devote our attention to the data structures based on hashing, the most relevant topic of this work.

### B.1    STATIC DATA STRUCTURES

The set membership problem is defined as follows. Let $[m]$ be the *universe*, and let the input be $S \subset [m]$ with $|S| = n$. Based on $S$, one writes data structure $f(S)$ to memory. Based only on $f(s)$, a query algorithm must be able to answer, for all $i \in [m]$, "Is $i \in S$?" using few probes to memory.

There are two models of probing:

1. Bit Probe Model (BPM): the *space* complexity is measured in the number of bits of $f(s)$ and the *query* complexity is measured in the number of bits accessed (probed);

2. Cell Probe Model (CPM): the data structure $f(S)$ is written in blocks, or *cells*, where each cell consists of $w$ bits; assume $w = \Omega(\log m)$; the *space* complexity is measured in the number of cells of $f(S)$ and the *query* complexity is measured in the number of cells accessed (probed).

Assume $m = n^{1+\Omega(1)}$. The smallest space complexity possible is $\log\binom{m}{n} \sim \Omega(n \log m)$ bits, or $\Omega(n)$ cells, as a data structure (together with a query algorithm) can be used to reconstruct $S$, and, therefore, must distinguish between all size $n$ subsets of $[m]$.

A simple solution in the CPM with optimal space complexity is to write all elements of $S$ to memory in sorted order. Then the query algorithm can answer a query for $i$ by doing binary search on the data structure. The query complexity is $O(\log n)$ cell probes (similar complexity is achieved in the dynamic setting by a binary search tree). The next subsection describes a data structure which works for the static case, in which the smallest memory possible is achieved and takes $O(1)$ probes in CPM.

## B.2 Set Membership in Linear Space and $O(1)$ Cell Probes

This subsection will present a data structure due to Fredman, Komlós, and Szemerédi Fredman et al. (1984) that supports set membership queries with constant query complexity and linear space in CPM. The data structure makes crucial use of 2-wise independent distributions. The complexity of the data structure is optimal up to constants: as argued, any data structure which answers all membership queries correctly needs to store $\Omega(n)$ cells (in the regime $m = n^{1+\Omega(1)}$) and needs to probe at least a single cell per query.

A natural strategy to solve the set membership problem is hashing: map the input to a set of $l$ cells using a one to one function. More precisely, the solution is supposed to find a family of functions $\mathcal{H} = \{h : [m] \to [l]\}$, where $l$ is as small as possible and for any $S \subseteq [m]$ there exists an $h \in \mathcal{H}$ such that the restriction $h|_S : S \to [l]$ of $h$ to $S$ is one to one. Given such a family, a data structure can be constructed as follows: write $S \cap h^{-1}(1)$ in the first cell, $S \cap h^{-1}(2)$ in the second, etc., and, $S \cap h^{-1}(l)$ in the $l$-th cell; also write down an identifier of the function $h$ which was used. Given a query element $i$, the query algorithm reads the identifier of $h$ and probes the $j$-th cell for $j = h(i)$. If that cell stores $i$, then $i$ is in $S$, otherwise it is not. Obviously, if $\mathcal{H}$ is the set of all functions from $m$ to $n$, then $\mathcal{H}$ satisfies the desired property: for any $S$ there is an $h \in \mathcal{H}$ which is one to one on $S$. However, such an $\mathcal{H}$ is too big: it has size $m^n$, so the optimal solution needs $\Omega(n)$ cells to identify a function $h \in \mathcal{H}$ and the query algorithm needs to probe $\Omega(n)$ cells to identify which $h$ was used. So it is also important to find a family $\mathcal{H}$ of polynomial size.

Allowing $l$ to be $n^2$, then 2-wise independence provides a family with the above properties. Let us first define what we mean by a pairwise independent hash family.

**Definition 3.** A pairwise-independent hash family is a set of functions $\mathcal{H} = h : [m] \to [l]$ such that for all $a, b \in [m]$ and all $c, d \in [l]$ we have $\Pr_h[h(a) = c \wedge h(b) = d] = 1/l^2$, where the probability is taken over choosing a uniformly random $h \in \mathcal{H}$.

Pairwise independent hashing is simply a different viewpoint on 2-wise independent distributions. When $h$ is chosen at random from $\mathcal{H}$, $h(1), \ldots, h(m)$ are $m$ 2-wise independent random variables taking values in $[l]$. Similarly, any $m$ 2-wise independent random variables $x_1, \ldots, x_m$ taking values in $[l]$ give a pairwise independent hash family: we can associate a hash function $h$ to every choice of random bits and set $h(a) = x_a$ for all $a \in [m]$. A pairwise-independent hash family can be contrasted as $\mathcal{H} = \{ax + b : a, b \in \mathbb{F}\}$ for a field $\mathbb{F}$ of size $l$. Such a family $\mathcal{H}$ has size $l^2$.

**A simple scheme.** Let us analyze the data structure described above when $\mathcal{H}$ is a pairwise independent family. First, we prove that a pairwise-independent $\mathcal{H}$ has a one to one function for each $S$.

**Lemma 4.** Let $\mathcal{H} = h : [m] \to [l]$ be a pairwise independent hash family and let $l \geq n^2$. Then for any $S \subseteq [m]$ of size $|S| = n$, there exists an $h \in \mathcal{H}$ such that the restriction $h|_S$ is a one to one function from $S$ to $[l]$.

*Proof.* Considering the probabilistic method, let us choose $h$ uniformly at random from $\mathcal{H}$. For $a, b \in S$, let $X_{a,b}$ be the indicator random variable that takes value 1 if and only if $h(a) = h(b)$. I.e.

$X_{a,b}$ is 1 if and only if $h$ makes $a$ and $b$ collide. The expected number of collisions will be

$$\mathbb{E}\left[\sum_{a<b} X_{a,b}\right] = \sum_{a<b} \mathbb{E}[X_{a,b}] = \frac{n(n-1)}{2l}. \tag{23}$$

In light of the fact that, by the pairwise independence of $\mathcal{H}$,

$$\Pr[X_{a,b} = 1] = \sum_{c\in[l]} \Pr[h(a) = h(b) = c] = \frac{1}{l}. \tag{24}$$

Since $l \geq n^2$, the expected number of collisions is less than $1/2$, and therefore there exists an $h^* \in \mathcal{H}$ that causes less than $1/2$ collisions of elements of $S$. But the number of collisions is an integer, therefore $h^*|_S$ is one to one. □

Lemma 4 gives a data structure with $O(n^2)$ cells and $O(1)$ query time: as discussed above, given $S$, we find an $h$ such that $h|_S$ is one to one, and we write down $h^{-1}(1) \cap S, \ldots, h^{-1}(l) \cap S$ as well as an identifier of $h$, for example $a$ and $b$ if we use $\mathcal{H} = ax + b$. Then a query algorithm reads the identifier of $h$ and, given a query $i$, reads $h^{-1}(j) \cap S$ where $h(i) = j$. The algorithms answers "yes" if and only if the result is $i$. Since the identifier of $h$ can be written in constant number of cells, and $h$ is one to one, evaluating a query takes constant time.

**The real scheme.** The next is going to present the full scheme of Fredman, Komlós, and Szemerédi Fredman et al. (1984). The disadvantage of the simple scheme above is that it uses suboptimal space: a quadratic number of cells, when only a linear number is necessary. As a first step, let us see how many collisions we have when we hash into a set of linear size.

**Lemma 5.** *Let $\mathcal{H} = h : [m] \to [l]$ be a pairwise independent hash family and let $l \geq n$. Then for any $S \subseteq [m]$ of size $|S| = n$, there exists an $h \in \mathcal{H}$ which causes less than $n/2$ elements of $S$ to collide.*

The proof of Lemma 5 is identical to the proof of Lemma 4. For input $S$, let us pick an $h$ that satisfies the conclusion of Lemma 5, i.e causes less than $n/2$ collisions of elements of $S$. Then in the data structure we write:

- for each $j \in [l]$:
  - the size of the inverse $n_j = |h^{-1}(j) \cap S|$;
  - a pointer to an instance of the simple scheme with input $h^{-1}(j) \cap S$;
- an identifier of $h$.

On query $i$, the query algorithm reads $h$ and reads the cells corresponding to $j = h(i)$; then it accesses the corresponding instance of the simple scheme and answers the query. Since the simple scheme supports answering queries with a constant number of probes, and the overhead to access the right instance of the simple scheme requires a constant number of probes, the query complexity is $O(1)$ probes. However, we need to analyze the space complexity of the data structure. Notice that the total number of cells used is up to a constant factor equal to $n + n_1^2 + \cdots + n_n^2$. The following lemma then completes the analysis.

**Lemma 6.** *Let $h : S \to [l]$ be such that there are at most $n/2$ collisions of elements of $S$. Then,*

$$\sum_{j=1}^{l} n_j^2 \leq 2n. \tag{25}$$

*Proof.* The number of elements $a, b$ that collide at $j$, i.e. $|\{(a, b) : a \neq b \wedge h(a) = h(b) = j\}|$, is $\binom{n_j}{2}$. Since the total number of collisions is assumed to be at most $n/2$, we have

$$\frac{n}{2} \geq \sum_{j=1}^{n} \binom{n_j}{2} = \sum_{j=1}^{n} \frac{n_j^2}{2} - \frac{n_j}{2} = \frac{1}{2}\sum_{j=1}^{l} n_j^2 - \frac{n}{2}. \tag{26}$$

The lemma follows after rearranging the terms. □

Putting everything together, we get the following theorem.

**Theorem 7.** *There exists a data structure for the set membership problem in the cell probe model that uses $O(n)$ cells of space and answers each membership query using $O(1)$ cell probes.*

## C    PROOF OF THEOREM 2

Recall that the estimation of $f_\theta(x_\nu)$ can be obtained using the Monte-Carlo method, i.e.,

$$\hat{f}_L(x_\nu) = \frac{|\Theta|}{L} \sum_{j=1}^{L} \alpha(w_j) g\left(w_j^{\mathrm{T}} \sum_{\mu \in \mathcal{N}(\nu)} \hat{A}_{\mu,\nu}^2 x_\mu\right), \tag{27}$$

where $\{w_1, w_2, \ldots, w_L\}$ is a sample of size $L$ selected randomly (and independently) from the uniform distribution with the probability measure

$$P(w) = \begin{cases} |\Theta|^{-1} & \text{if } w_j \in \Theta, \\ 0 & \text{otherwise.} \end{cases} \tag{28}$$

The concern of the approximation upper bound is to verify whether $d_V\left(f, \hat{f}_L\right)$ can be sufficiently small under certain conditions (e.g, L is sufficiently large). This can be answered by considering two parts, i.e., $d_V\left(f_\theta, \hat{f}_L\right)$ and $d_V(f, f_\theta)$, respectively, as we detail in the following. (i) For $d_V\left(f_\theta, \hat{f}_L\right)$: We first need to prove $E\left[\hat{f}_L(x_\nu)\right] = f_\theta(x_v)$. By Eq.27 one can obtain that

$$E\left[\hat{f}_L(x_\nu)\right] = E\left[\frac{|\Theta|}{L} \sum_{j=1}^{L} \alpha(w_j) g\left(w_j^{\mathrm{T}} \sum_{\mu \in \mathcal{N}(\nu)} \hat{A}_{\mu,\nu}^k x_\mu\right)\right]. \tag{29}$$

Since the $w_j(k = 1, 2, \ldots, L)$ are drawn independently from the probability distribution (Eq.28), this gives

$$\begin{aligned} E\left[\hat{f}_L(x_\nu)\right] &= E\left[\alpha(w) g\left(w^T \sum_{\mu \in \mathcal{N}(\nu)} \hat{A}_{\mu,\nu}^k x_\mu\right)\right] \\ &= \int |\Theta|\alpha(w) g\left(w^T \sum_{\mu \in \mathcal{N}(\nu)} \hat{A}_{\mu,\nu}^k x_\mu\right) P(w) dw \\ &= \int_\Theta |\Theta|\alpha(w) g\left(w^T \sum_{\mu \in \mathcal{N}(\nu)} \hat{A}_{\mu,\nu}^k x_\mu\right) \frac{1}{|\Theta|} dw \\ &= f_\theta(x_\nu). \end{aligned} \tag{30}$$

In the same way, the variance is obtained as

$$
\begin{aligned}
\mathrm{Var}\left[\hat{f}_L\left(x_\nu\right)\right] &= \mathrm{Var}\left[\frac{|\Theta|}{L}\sum_{j=1}^{L}\alpha\left(w_j\right)g\left(w_j^T\sum_{\mu\in\mathcal{N}(\nu)}\hat{A}^k_{\mu,\nu}x_\mu\right)\right]\\
&=\frac{1}{L}\mathrm{Var}\left[\alpha(w)|\Theta|g\left(w^T\sum_{\mu\in\mathcal{N}(\nu)}\hat{A}^k_{\mu,\nu}x_\mu\right)\right]\\
&=\frac{1}{L}E\left[\left(\alpha(w)|\Theta|g\left(w^T\sum_{\mu\in\mathcal{N}(\nu)}\hat{A}^k_{\mu,\nu}x_\mu\right)\right)^2\right]-\frac{1}{L}E\left[\alpha(w)|\Theta|g\left(w^T\sum_{\mu\in\mathcal{N}(\nu)}\hat{A}^k_{\mu,\nu}x_\mu\right)\right]^2\\
&=\frac{1}{L}\left[\left(\alpha(w)|\Theta|g\left(w^T\sum_{\mu\in\mathcal{N}(\nu)}\hat{A}^k_{\mu,\nu}x_\mu\right)\right)^2\right]-\frac{1}{L}\left(f_\theta\left(x_\nu\right)\right)^2\\
&=\frac{1}{L}\int\left(\alpha(w)|\Theta|g\left(w^T\sum_{\mu\in\mathcal{N}(\nu)}\hat{A}^k_{\mu,\nu}x_\mu\right)\right)^2 P(w)dw-\frac{1}{L}\left(f_\theta\left(x_\nu\right)\right)^2\\
&=\frac{|\Theta|}{L}\int_\Theta\left(\alpha(w)g\left(w^T\sum_{\mu\in\mathcal{N}(\nu)}\hat{A}^k_{\mu,\nu}x_\mu\right)\right)^2 dw-\frac{1}{L}\left(f_\theta\left(x_\nu\right)\right)^2.
\end{aligned}
\tag{31}
$$

So far we can have

$$
\begin{aligned}
d_V^2\left(f_\theta,\hat{f}_L\right) &= \frac{1}{|V|}E\left[\sum_\nu\left(f_\theta\left(x_\nu\right)-\hat{f}_L\left(x_\nu\right)\right)^2\right]\\
&=\frac{1}{|V|}\sum_\nu E\left[\left(f_\theta\left(x_\nu\right)-\hat{f}_L\left(x_\nu\right)\right)^2\right]\\
&=\frac{1}{|V|}\sum_\nu E\left[\left(E\left[\hat{f}_L\left(x_\nu\right)\right]-\hat{f}_L\left(x_\nu\right)\right)^2\right]\\
&=\frac{1}{|V|}\sum_\nu \mathrm{Var}\left[\hat{f}_L\left(x_\nu\right)\right].
\end{aligned}
\tag{32}
$$

We can now substitute the expression for the variance, i.e. Eq.31, to obtain

$$
d_V^2=\left(f_\theta,\hat{f}_L\right)\frac{1}{L|V|}\sum_\nu\left(\int_\Theta|\Theta|\left(\alpha(w)g\left(w^T\sum_{\mu\in\mathcal{N}(\nu)}\hat{A}^k_{\mu,\nu}x_\mu\right)\right)^2 dw-\left(f_\theta\left(x_\nu\right)\right)^2\right).
\tag{33}
$$

Since $|g(x)|\le 1$, the above can be bounded by

$$
d_V^2\left(f_\theta,\hat{f}_L\right)\le\frac{1}{L|V|}\sum_\nu\left(|\Theta|\int_\Theta\alpha^2(w)dw\right)=\frac{|\Theta||V|}{L|V|}\int_\Theta\alpha^2(w)dw=\frac{|\Theta|}{L}\int_\Theta\alpha^2(w)dw.
\tag{34}
$$

This gives

$$
d_K\left(f_\theta,\hat{f}_L\right)\le\frac{C}{\sqrt{L}},
\tag{35}
$$

where

$$
C:=\sqrt{|\Theta|\int_\Theta\alpha^2(w)dw}.
\tag{36}
$$

(ii) For $d_V\left(f, f_\theta\right)$: It seems that $d_V\left(f, f_\theta\right)$ can be made arbitrarily small by choosing large enough values for parameter $\theta$. However, this conflicts with the previously derived $d_K\left(f_\theta, \hat{f}_L\right)$ as it can be very large when $\theta$ goes to infinity. This means the setting of $\theta$, which directly determines the uniform distribution for the random parameters, should be chosen properly for a given modeling task since the functional class and complexity of the unknown function cannot be prejudged. Here for our theoretical interpretation, we provide two kinds of assumptions (AS1 and AS2 below) to restrict the class of continuous functions f to be smoother such that $d_V\left(f, f_\theta\right)$ can be sufficiently small for finite $\theta$. (AS1) Suppose that the to-be-approximated function defined on the graph has the integral representation

$$f\left(x_\nu\right) = \int_\Omega \alpha(w) g\left(w^T \sum_{\mu \in \mathcal{N}(\nu)} \hat{A}_{\mu, \nu}^k x_\mu\right) dw, \tag{37}$$

where $\Omega \subset R^m$ is a compact space. Then, there exists $\Theta^* \asymp \Omega$ such that $d_V\left(f, f_\theta\right) \leq \epsilon$, where $\epsilon$ is sufficiently small and negligible. Similar results from the statistical learning theory perspective can be found in Rahimi & Recht (2008).

(AS2) Based on the similar assumption mentioned in Igelnik & Pao (1995), one can restrict the class of continuous functions f to satisfy the Lipshitz condition Adams & Fournier (2003) (Note that the formulism of the condition is still applicable for continuous functions defined on the graph vertex domain), and simultaneously, restrict the support of the activation function $g$ to the interval with length $\delta_L \theta$, where $\lim_{L \to +\infty} \delta_L = 0$, then $d_V\left(f, f_\theta\right) \leq \epsilon$ can be verified. See the proof of Theorem 3 in Igelnik & Pao (1995). According to (i) and (ii), we can conclude that under mild conditions

$$d_V\left(f, \hat{f}_L\right) \leq \frac{C}{\sqrt{L}}, C := \sqrt{|2\theta|^m \int_\Theta \alpha^2(w) dw}. \tag{38}$$

