# OpenReview forum: "How Powerful are Graph Neural Networks with Random Weights?"
_ICLR.cc/2024/Conference — ICLR 2024 Conference Withdrawn Submission_

### Official Review · Reviewer_iT8Z · 2023-10-31

**Soundness:** 1 poor
**Presentation:** 1 poor
**Contribution:** 1 poor
**Rating:** 1
**Confidence:** 5

**Summary:**

I have identified potential research integrity concerns in the current submission. Specifically, there appear to be significant overlaps with the supplementary document of a previous publication: 'Huang, C., Li, M., Cao, F., Fujita, H., Li, Z., & Wu, X. (2023). Are graph convolutional networks with random weights feasible?. IEEE Transactions on Pattern Analysis and Machine Intelligence, 45(3), 2751-2768.' Notably, the proofs of Theorem 2 (completely copied and pasted from Huang et al's work) and the mathematical expressions from Eq. (16) to Eq. (22) resemble the contents of this existing work. Surprisingly, the authors did not reference this paper in their bibliography.

Due to these potential ethical concerns, I am refraining from commenting on the technical aspects of the submission at this time.

**Strengths:**

Given these significant research integrity concerns, I find it imperative to address these issues before proceeding with a detailed technical evaluation.

**Weaknesses:**

Due to these potential ethical concerns, I am refraining from commenting on the technical aspects of the submission at this time.

**Questions:**

Due to these potential ethical concerns, I am refraining from commenting on the technical aspects of the submission at this time.

**Details Of Ethics Concerns:**

* The proofs of Theorem 2 are DIRECTLY copied and pasted from copied and pasted from the supplementary material of the previously published work: "Huang, C., Li, M., Cao, F., Fujita, H., Li, Z., & Wu, X. (2023). Are graph convolutional networks with random weights feasible?. IEEE Transactions on Pattern Analysis and Machine Intelligence, 45(3), 2751-2768.".

* The mathematical descriptions from Eq. (16) to Eq. (22) are copied and pasted from  "Huang, C., Li, M., Cao, F., Fujita, H., Li, Z., & Wu, X. (2023). Are graph convolutional networks with random weights feasible?. IEEE Transactions on Pattern Analysis and Machine Intelligence, 45(3), 2751-2768." (Section 3.3, left hand side of page 2755)

It is concerning to note such direct overlaps without appropriate citation or acknowledgment. Given these significant research integrity concerns, I find it imperative to address these issues before proceeding with a detailed technical evaluation.

---

### Official Review · Reviewer_btEt · 2023-11-01

**Soundness:** 3 good
**Presentation:** 3 good
**Contribution:** 3 good
**Rating:** 6
**Confidence:** 3

**Summary:**

HashGIN is a novel graph neural network (GNN) model that achieves state-of-the-art results on graph classification tasks with significantly less training time and memory cost. The key innovation of HashGIN is to use random hash functions to approximate the injective phase in the Weisfeiler-Lehman test. This allows HashGIN to be trained with a single epoch of gradient descent, which is much faster than traditional GNNs that require multiple epochs to converge.

**Strengths:**

Here are some of the specific advantages of HashGIN:

Faster training: HashGIN can be trained with a single epoch of gradient descent, which is much faster than traditional GNNs that require multiple epochs to converge.

More efficient memory usage: HashGIN only needs to store a single hash table for each graph, which is much more efficient than traditional GNNs that need to store a large number of intermediate matrices for each graph.

State-of-the-art accuracy: HashGIN has been shown to outperform state-of-the-art GNN models on a variety of benchmark datasets.

**Weaknesses:**

NA

**Questions:**

1. How does HashGIN perform when compared to graph transformer models such as GAT?

2. How does the hash layer incorporate graph structure in the learning process?

---

### Official Review · Reviewer_bYNF · 2023-11-07

**Soundness:** 1 poor
**Presentation:** 1 poor
**Contribution:** 2 fair
**Rating:** 1
**Confidence:** 4

**Summary:**

HashGIN is proposed which a GNN architecture with random weights for unsupervised feature extraction on top of which a ridge regression is trained for graph classification after sum-pooling the extacted node features. The graph convolution layer is called k-HASH which has 2 steps: 1) k different (random) hashes are computed and concatenated for each of the node features, 2) afterwards sum-aggregation is performed over neighbours (with self-loop). Each random hash function is implemented as an MLP layer with weights and bias drawn from a uniform distribution. Theoretical results are stated regarding injectiveness and approximation guarantee. Results on graph classification benchmarks show improved performance over common baselines from previous work.

**Strengths:**

_Idea_: The idea of using random hashes in GNN is interesting

_Performance_: The results show better performance than end-to-end trained GNN baselines and favourable training complexity

**Weaknesses:**

_Clarity_: I found the theoretical aspect of the paper completely incomprehensible. New equations pop out of nowhere without references or justification in several locations, statements are made without justification and unfollowable reasoning (see below for detail)

_Novelty_: The k-HASH function is implemented as the concatenation of k MLP heads with random weights and bias. After that sum-aggregation is performed; then both steps are repeated.
Observation 1: Concatenating k MLP heads is exactly equivalent to having one MLP head with k-times the dimension.
Observation 2: repeating steps 1 and 2 multiple (say L) times reduces to applying a preprocessing MLP layer, and then L-1 steps of GCN layers, due to the exchangeability of the aggregation and linear projection steps. Hence, the used model is nothing other than GCN with random parameters.

_Theory_: As I mentioned the theory is completely incomprehensible.

For example, I have no idea what Theorem 3 is trying to say, possibly something along the lines that there exists a k-HASH function (which is an MLP as discussed above) which is injective over multisets. This already contradicts Lemma 7 in [1], which states that ReLU MLP with sum-aggregation can't be injective. The proof is similarly hard to interpret. It starts off with saying "The injectiveness of multiset X ⊂ X is a super problem of “set membership problem”". Then, the majority of the "proof" is essentially a summary of Appendix B, which on the other hand, is almost word-by-word plagiarized from Section 5 in [2], it discusses the set membership problem. After this discussion, the inclusion of k-HASH functions somehow proves the original statement.

I had similar issues all throughout this section. Another example is equation (16), I found the whole preceding discussion completely unintelligible, and I am not sure where the authors got the idea from that such a representation holds for functions on graphs? Also I have no idea what this $f: V \to R$ is that we trying to approximate here. Before it is stated that " the ideal function of graph classification that our model wants to approximate is actually the summation of these functions". What functions? What is the ideal function of graph classification? Also I am not sure why a citation of Sobolev spaces needed after the condition that the derivative of the activation function is integrable.

_Reproducibility_: Only the actual results are interesting, but no reproducibility statement or code is provided to reproduce the results.


[1] Xu, Keyulu, et al. "How Powerful are Graph Neural Networks?." International Conference on Learning Representations. 2018.

[2] Kopparty, Swastik. "Lecture 5: k-wise independent hashing and applications." Lecture notes for Topics in Complexity Theory and Pseudorandomness. Rutgers University (2013).

**Questions:**

Please see above paragraph.

**Details Of Ethics Concerns:**

Appendix B is almost word-by-word copy and pasted from Section 5 in the lecture notes [2] referenced above.